# An Emerging Syndemic of Smoking and Cardiopulmonary Diseases in People Living with HIV in Africa

**DOI:** 10.3390/ijerph18063111

**Published:** 2021-03-18

**Authors:** Emmanuel Peprah, Mari Armstrong-Hough, Stephanie H. Cook, Barbara Mukasa, Jacquelyn Y. Taylor, Huichun Xu, Linda Chang, Joyce Gyamfi, Nessa Ryan, Temitope Ojo, Anya Snyder, Juliet Iwelunmor, Oliver Ezechi, Conrad Iyegbe, Paul O’Reilly, Andre Pascal Kengne

**Affiliations:** 1School of Global Public Health, New York University, New York, NY 10012, USA; mah842@nyu.edu (M.A.-H.); sc5810@nyu.edu (S.H.C.); gyamfj01@nyu.edu (J.G.); ryann01@nyu.edu (N.R.); to790@nyu.edu (T.O.); as13565@nyu.edu (A.S.); 2Mildmay Uganda, Kampala, Uganda; barbara.mukasa@mildmay.or.ug; 3School of Nursing, Columbia University, New York, NY 10032, USA; jyt2116@cumc.columbia.edu; 4Division of Endocrinology, Diabetes and Nutrition, Department of Medicine, University of Maryland School of Medicine, Baltimore, MD 21201, USA; hxu1@medicine.umaryland.edu; 5Department of Neurology, University of Maryland School of Medicine, Baltimore, MD 21201, USA; linda.chang@umm.edu; 6College for Public Health and Social Justice, Saint Louis University, St. Louis, MO 63103, USA; juliet.iwelunmor@slu.edu; 7Nigerian Institute of Medical Research, Lagos, Nigeria; oliverezechi@yahoo.co.uk; 8Icahn School of Medicine at Mount Sinai, Mount Sinai Hospital, New York, NY 10029, USA; conrad.1.iyegbe@kcl.ac.uk (C.I.); paul.oreilly@mssm.edu (P.O.); andre.kengne@mrc.ac.za (A.P.K.); 9South African Medical Research Council, Cape Town, South Africa

**Keywords:** syndemics, cardiovascular disease, smoking, tobacco, low- and middle-income countries, HIV/AIDS

## Abstract

Background: African countries have the highest number of people living with HIV (PWH). The continent is home to 12% of the global population, but accounts for 71% of PWH globally. Antiretroviral therapy has played an important role in the reduction of the morbidity and mortality rates for HIV, which necessitates increased surveillance of the threats from pernicious risks to which PWH who live longer remain exposed. This includes cardiopulmonary comorbidities, which pose significant public health and economic challenges. A significant contributor to the cardiopulmonary comorbidities is tobacco smoking. Indeed, globally, PWH have a 2–4-fold higher utilization of tobacco compared to the general population, leading to endothelial dysfunction and atherogenesis that result in cardiopulmonary diseases, such as chronic obstructive pulmonary disease and coronary artery disease. In the context of PWH, we discuss (1) the current trends in cigarette smoking and (2) the lack of geographically relevant data on the cardiopulmonary conditions associated with smoking; we then review (3) the current evidence on chronic inflammation induced by smoking and the potential pathways for cardiopulmonary disease and (4) the multifactorial nature of the syndemic of smoking, HIV, and cardiopulmonary diseases. This commentary calls for a major, multi-setting cohort study using a syndemics framework to assess cardiopulmonary disease outcomes among PWH who smoke. Conclusion: We call for a parallel program of implementation research to promote the adoption of evidence-based interventions, which could improve health outcomes for PWH with cardiopulmonary diseases and address the health inequities experienced by PWH in African countries.

## 1. Introduction

HIV continues to be a major global public health threat. While new infections and the number of people living with HIV/AIDS (PWH) have declined from its peak in 2005, this decline has not been as dramatic in resource-constrained settings as in high-income countries (HICs) [1]. The Joint United Nations Programme on HIV/AIDS (UNAIDS) indicates that ~1.7 million individuals worldwide acquired HIV in 2019, marking a 23% decline in new HIV infections since 2010 [2]. Although the decreases in new infections are significant, currently, 38.0 million (31.6–44.5 million) people globally are living with HIV. 

Africa is home to approximately 12% of the global population, and 71% of the population are disproportionately infected with HIV [1]. The African countries with the highest populations of PWH within Africa include South Africa at 25%; Nigeria at 13%; Uganda, Mozambique, Tanzania, Zimbabwe, Kenya, each at 6%; Zambia and Malawi at 4%; and Ethiopia at 3%. Together, this comprises > 75% of all people living with HIV [1]. The epidemic is particularly concentrated in Nigeria, South Africa, and Uganda; UNAIDS recorded 1.9 million (0.07% incidence), 7.7 million (0.49%), and 1.4 million (0.14%) seropositive individuals in these countries, respectively (Table 1) [3]. Moreover, five in six new infections in Africa occur among adolescents aged 15–19 years [2]. The data suggest that the epidemic has shifted to a younger demographic from the previous high-risk group being 15–24 years. Africa has a population of 1.4 billion people with a median age of 18 years old; this is a larger population of young adults compared to most high-income global economic regions [4]. There are concerns that within a few years, this demographic shift could make the last 30 years of progress from preventing and eradicating the spread of HIV unsustainable [5]. 

**Table 1 ijerph-18-03111-t001:** Prevalence data for PWH, smoking, and the general adult populations.

Country	Official Population [6]	PWH [7]	Smoking Prevalence among Adults (Non-PWH)	Smoking Prevalence among PWH
Nigeria	214 million	1.9 million	10.40% [8]	22.10% [8]
South Africa	56.5 million	7.7 million	17.60% [9]	24.88% [10]
Uganda	43.3 million	1.4 million	7.40% [11]	10.00% [12]

PWH = people living with HIV/AIDS.

In response to the disproportionately high number of PWH, national programs have been implemented to decrease the infection rate and to increase access to combination antiretroviral therapy (cART). Although these programs have had remarkable success in Africa, the number of people on cART fails to meet the “90–90–90” goal established by UNAIDS to significantly curtail the spread of HIV worldwide [13,14,15]. The 90–90–90 goal projects that by 2020 (i) 90% of all people with HIV should know their HIV status, (ii) 90% of all people who know their status should be on cART, and (iii) 90% of all people receiving cART should have achieved viral suppression [16]. Despite the varying levels of success, none of the three high-burdened countries have reached these goals. Moreover, the variability in the data to estimate the incidence rates for PWH in different countries highlights the challenges and failures of characterizing and implementing broad transnational intervention programs to reduce the burden of HIV. Furthermore, tailored programs to address the evolving nature of the HIV epidemic are needed to achieve health equity [15,17].

### 1.1. Smoking Prevalence Is Higher among PWH in Nigeria, South Africa, and Uganda and Is Associated with Negative Health Outcomes

Globally, tobacco cigarette smoking kills 8 million people per year. In contrast to high-income countries (HICs), the smoking prevalence is increasing in low- and middle-income countries (LMICs), accounting for the majority of smoking-related deaths [18,19]. Cross-sectional studies examining tobacco use show 2–4-fold higher tobacco utilization among PWH compared to the general population (Table 1) [20]. For example, in Nigeria, 10% of the general population smoke compared to 22% of PWH [8,21]. In South Africa, the smoking prevalence is 18% in the general population but 25% among PWH [9,10]. In Uganda, 7.4% of the general population smoke compared to 10% of PWH [11,12]. Findings from a cross-sectional survey of 25 sub-Saharan African countries suggest that living with HIV/AIDS is a strong predictor of smoking in men and women, even after adjustment for demographic, socioeconomic, and other factors [22]. The burden of tobacco-related illness among PWH may be even greater in resource-limited settings that cannot match the spending of HICs on evidence-based interventions that target smoking cessation. 

The negative health impacts of tobacco smoking are well established; smoking is associated with poorer quality of life as measured by lower scores for general health perception, physical functioning, bodily pain, energy, and cognitive functioning for PWH [23,24]. Tobacco smokers are at increased risk for numerous illnesses, including cancer, cardiovascular diseases (CVDs), coronary artery disease (CAD), chronic obstructive pulmonary disease (COPD), stroke, and diabetes [1,3]. This culminates in increased odds of premature death compared with non-smokers [25]. Elevated rates of smoking in PWH also raise concerns about outcomes, which are endemic in PWH and may be exacerbated by smoking. This includes opportunistic infections, such as bacterial pneumonias, acute bronchitis, and tuberculosis, higher viral load, and lower CD4 cell counts. Other complications may arise as a direct result of antiretroviral medications themselves [26,27,28]. Suppression of local lung defenses is a common etiologic pathway that may explain higher rates of pulmonary illness in HIV [24,29]. An increased incidence of cancers (e.g., lung and cervical cancers) has been documented among tobacco-using PWH relative to PWH that do not smoke. At the cellular level, PWH who smoke show evidence of accelerated cellular aging, which may explain the younger age of onset of cancer and cardiopulmonary disease in this group [30,31,32]. Preventing tobacco-related illnesses and death among PWH requires a multifaceted approach to discourage smoking to reduce morbidity and mortality. 

### 1.2. Prevalence of Hypertension, COPD, and CAD Could Be Elevated among PWH Who Smoke, but Data Are Lacking

Smoking-related comorbidities of HIV, including cardiovascular risk factors, such as hypertension, COPD, and CVDs (e.g., CAD), are increasing in African populations [33,34,35,36]. Incomplete epidemiological transition from communicable to non-communicable/chronic diseases with concentrated populations of PWH in sub-Saharan Africa (SSA) has produced a double burden of disease in many African settings [37]. 

Hypertension is a major risk factor for CVDs that disproportionately affects populations in Africa [38,39,40,41]. Studies that have examined hypertension prevalence among PWH show increased prevalence in Nigeria, with similar rates of hypertension in South Africa, but lower prevalence in Uganda compared to the general population. For example, in Nigeria, 28.9% of the general population are hypertensive compared to 46% of PWH [42,43]. In South Africa and Uganda, hypertension prevalence is >40% and 26.40%, respectively, in the general population, but ~39% and 10%–11% among PWH [44,45,46,47]. The increasing prevalence of CVD risk factors among the general population, the aging of PWH, and the increased chronic inflammation experienced by PWH suggest that CVDs are likely to become more common among PWH [48]. 

Smoking is a well-established primary risk factor for COPD among PWH [12,21,29,49,50,51,52,53,54,55,56]. South Africa has the largest population of PWH who also smoke; the high prevalence of smoking among PWH contributes to higher rates of COPD among seropositive individuals (Table 2). The greater smoking prevalence among PWH has been linked to higher COPD and hypertension prevalence among PWH compared to uninfected individuals in Uganda and Nigeria, although robust data are presently lacking [8,14,15,16,17,18,19,20].

**Table 2 ijerph-18-03111-t002:** Prevalence data for cardiopulmonary disease, including coronary artery disease (CAD), chronic obstructive pulmonary disease (COPD), and hypertension (HTN), among the general population and PWH (non-smokers) in Nigeria, South Africa, and Uganda.

Country	CAD Prevalence among Adults (Non-PWH)	CAD Prevalence PWH	COPD Prevalence among Adults (Non-PWH)	COPD Prevalence PWH	HTN Prevalence among Adults (Non-PWH)	HTN Prevalence PWH
Nigeria	1.6% to 3.4% [57,58]	NA	7.70% [52]	22.19% to 15.4% [51]	28.90% [42]	46.0% [43]
South Africa	11% [58]	NA	16.7% to 22.20% [59]	8.00% to 9.8% [60,61]	>40% [45]	38.6% [44]
Uganda	4.02% [62]	NA	6.20% [63]	3.7% to 10.4% [64]	26.40% [54]	10% to 11% [46,47]

PWH = people living with HIV/AIDS.

Studies from high-income settings have observed an increased incidence of CAD among PWH who are smokers compared to the general population of smokers [65,66,67]. However, no data are available on CAD for PWH in Africa (Table 2), and observations from settings such as Denmark and the United States may be a poor proxy for estimating the incidence of CAD in African populations. Although statistical models indicate that CAD is the leading cause of CVD mortality in SSA, followed by stroke and hypertensive heart disease, real field data suggest that the rates are still relatively low compared to high-income countries [68]. CAD accounts for only 12% of the CVD disability-adjusted life years (DALYs) in sub-Saharan Africa, compared with 51% of DALYs in HICs [68]. While the prevalence of CAD and myocardial infractions (MIs) is considered low in Africa compared to HICs, the true prevalence of CAD in the region may be underestimated in epidemiological models due to systematic failure to diagnose CAD cases that result in premature mortality for PWH [69,70]. The lack of robust studies in African populations significantly limits the understanding of CAD in Africa. Further, epidemiological models indicate that CVDs, including CAD, will continue to increase in Africa [57,71,72]. This increase may be particularly prominent among PWH, given their higher rates of smoking. 

### 1.3. Inflammation Underlies Cardiopulmonary Diseases in PWH

Epidemiological evidence establishes correlations between smoking, hypertension, CVDs, and COPD in PWH with poor health outcomes. Cohort studies of PWH in the United States highlight greater mortality from CVDs and non-AIDS malignancies in smokers than in non-smokers [73,74]. PWH who smoke have elevated plasma inflammatory biomarkers that strongly predict cardiovascular events and all-cause mortality [30,67,75]. Several studies have found associations between the markers of inflammation, coagulation, and HIV mortality; these include interleukin 6 (IL-6), highly sensitive CRP (hsCRP), sCD14, and d-dimers [76,77,78,79]. 

Moreover, elevated plasma inflammatory markers (e.g., IL-6 and tumor necrosis factor (TNF)), have been reported in individuals with higher levels of markers of microbial translocation [67]. Microbial translocation is an early feature of HIV infection and occurs when impairment of the gut barrier leads to a process whereby microbial products leak through the intestinal barrier and cause immune activation [67]. For example, the *Prevotella* spp. present in gut microbiota, when translocated to the interstitial space, can release lipopolysaccharides that can initiate significant immune responses via binding CD14, found either in soluble form or anchored on the surface of monocytes and macrophages [80]. The newly formed complex LPS/CD14 can activate Toll-like receptor-4 (TLR4), leading to the production of pro-inflammatory cytokines (e.g., IL-16) in PWH [80,81]. Hsue and colleagues speculated that microbial translocation is another mechanism that might contribute to the atherogenesis associated with HIV infection via chronic inflammation, which induces endothelial dysfunction and vascular inflammation, resulting in atherosclerotic cardiovascular disease (ASCVD) [67]. 

Within PWH, chronic inflammation from microbial translocation can be further exacerbated by smoking, which can also produce inflammation in lung alveolar cells [82]. Indeed, a recent mechanistic review highlighted that exposure to tobacco smoke leads to increased mucosal inflammation and increased expression of inflammatory cytokines (e.g., IL-8, IL-6, and TNF) [82]. Cigarette smoking can induce alveolar inflammation, leading to COPD in PWH that involves multiple injurious processes, resulting in inflammation. Receptor-mediated signal transduction pathways activated in response to reactive oxygen species and tobacco components lead to chronic airway responses, atherogenesis, and alveolar destruction [83,84]. The complex inflammatory processes induced by cigarette smoking and microbial translocation occurring in PWH share several overlapping key cytokines/chemokines (e.g., IL-6 and TNF), but the mechanistic contributions of each pathway to the inflammatory process in PWH who smoke are unknown. Our preliminary analysis from a sub-cohort of PWH from a community-based sample of Africans from Soweto and South Africa, including 103 HIV-infected men (63 smokers and 40 non-smokers), found significant upregulation of several inflammatory serum markers in smokers compared with non-smokers (unpublished data). Some of these biomarkers included several interleukins and other pro-inflammatory proteins. Many of these proteins heavily interact with one another to promote inflammation synergistically downstream, which could suggest that in atherosclerotic cardiovascular disease, one potential signaling mechanism could be the pro-inflammatory JAK–STAT signaling pathway involving macrophages and TNFα production [85]. 

The current data on signaling pathways for PWH who smoke are incomplete because many underlying mechanisms of smoking-induced inflammatory diseases are not yet known [86]. However, several reports suggest that smoking activates the NLRP3 inflammasome in the lung epithelium, with increased expression of pro-IL-1β and caspase-1, as well as cytokines IL-1β and IL-18, in smokers with COPD [87,88,89]. Another study found that cigarette smoke extract inhibits NLRP3 and also activates caspase 1 via an NLRP3-independent and TLR4–TRIF–caspase-8 axis [90]. These observations are contradictory, suggesting that the inflammatory state could be mediated by inflammasome-related genes or a non-inflammasome pathway. Thus, more research is needed to discern whether smoking leads to inflammasome NLRP3 or TLR4–TRIF pathways in mediating the inflammation in PWH [91]. 

PWH aged 35 years have a median life expectancy of 62.6 years (95% CI, 59.9–64.6 years) if a smoker and 78.4 years (95% CI, 70.8–84.0 years) otherwise. Additional life years are lost due to smoking than to HIV (12.3 versus 5.1); the population-attributable risk of death due to smoking is 61.5% in PWH and 34.2% in people without HIV [67,92]. The exacerbated risk due to smoking in PWH may suggest unique molecular mechanisms related to poor health outcomes that may only be present in PWH but not in people without HIV. Context-specific research in the adverse effect of smoking in PWH is warranted even with the existing research of smoking in general non-HIV populations. Molecular evidence suggests cigarette smoke-induced inflammation halts the proliferation of lung fibroblasts and upregulates two pathways linked to cell senescence [93]. Moreover, epidemiological studies highlight that the all-cause mortality for PWH who smoke is much higher than in non-smokers, and PWH who smoke have worse health outcomes. PWH who smoke experience accelerated cellular aging and greater inflammation, both from smoking and microbial translocation, which is associated with cellular senescence and can therefore elicit an earlier onset of disease. Moreover, the population-attributable risk of smoking in PWH can be quantified based on current studies that stratify phenotypes for heart failure in PWH non-smokers using biomarkers [94]. For example, Scherzer et al. used eight serum biomarkers (e.g., ST2, NT-proBNP (N-terminal pro-B-type natriuretic peptide), hsCRP, GDF-15 (growth differentiation factor 15), cystatin C, IL-6, D-dimer, and troponin) to derive cardiac phenotypes for PWH non-smokers [94]. Three clusters were included: Cluster 3, which had higher levels of CRP, IL-6, and D-dimer (“inflammatory phenotype”) and was associated with a 51% increased risk of diastolic dysfunction; Cluster 2, which displayed elevated levels of ST2, NT-proBNP, and GDF-15 (“cardiac phenotype”) and was associated with a 67% increased risk of pulmonary hypertension (defined as echocardiographic PASP ≥ 35 mmHg); and Cluster 1, which had lower levels of both phenotype-associated biomarkers [94]. Although PWH who smoke were not included in their analysis, this approach could be used to improve clinical management and to generate significant knowledge on the early onset of atherosclerotic cardiovascular disease.

As noted by Hsue and Waters, although research on the inflammatory biomarkers of the risk of CVDs in HIV infection has been performed, additional work is needed to identify the best biomarker in the setting of HIV infection, along with the effect of HIV-related factors, traditional CVD risk factors, and smoking [67]. To elucidate the biological pathways that underlie smoking, inflammation, and cardiopulmonary diseases in PWH, a novel transdisciplinary approach bridging genomics and social–behavioral sciences is needed to interrogate both the biological and social–behavioral pathways that put PWH who are smokers at elevated risk. Large-scale prospective cohort studies provide significant insights into key lifestyle, environmental, molecular, and genomic determinants of the health outcomes among PWH. However, there is currently no such cohort study of cardiopulmonary disease among PWH in Africa who smoke; a large, multi-modal cohort study is urgently needed to fill these knowledge gaps.

### 1.4. The Syndemic Burden for PWH and the Economic Impact of Not Addressing Comorbid Conditions

Advances in the treatment regimens for HIV have significantly increased the life expectancy for PWH. Widespread testing and diagnosis, with linkage to follow-up healthcare (e.g., cART initiation and retention) via established HIV treatment cascades, have enabled low and undetectable viral loads [15,55]. Undetectable viral loads, in turn, effectively disrupt the transmission of HIV [95,96]. cART utilization has resulted in significant reductions in morbidity and mortality. However, as the lifespan of PWH increases, so increases the opportunity for a variety of HIV-related comorbidities.

As PWH age, the clustering of HIV, COPD, and CVDs with linkages to smoking calls for smoking cessation programs specific to PWH. However, the mental, environmental, and economic circumstances that predispose PWH to smoke at high rates compared to the general population cannot be addressed solely with smoking cessation programs. We therefore advocate for both biomechanistic and implementation research to strengthen prevention and care strategies by considering the full scope of vulnerabilities, rather than treating the disorders individually and ignoring the complex contexts in which they occur (Figure 1) [97]. Syndemics, or “synergistic epidemics,” describes the complex interaction between two or more epidemics with consideration for the multilevel social and environmental context in which they arise [98]. A syndemic framework can help elucidate the synergistic effects of each co-occurring morbidity more astutely than an independent examination of each. The causality between synergistic epidemics is not unidirectional, but rather multidirectional, and contributes to an overall greater burden of disease [99]. Although the syndemic framework addresses the adverse impact of environmental and social factors on disease clusters, highlighting how social environments in particular contribute to the conditions of social inequality and injustice that shape disease clustering, disease interaction, and vulnerability to diseases [100], it does not provide a focus on the implementation of evidence-based interventions to address these injustices and to achieve health equity. By utilizing implementation research in the context of a syndemic framework, interventions may be bundled to target co-occurring epidemics together, thereby potentially increasing the reach to marginalized populations, reducing the cost, and ensuring the efficient use of resources, which is critical in the resource-constrained settings of LMICs.

**Figure 1 ijerph-18-03111-f001:**
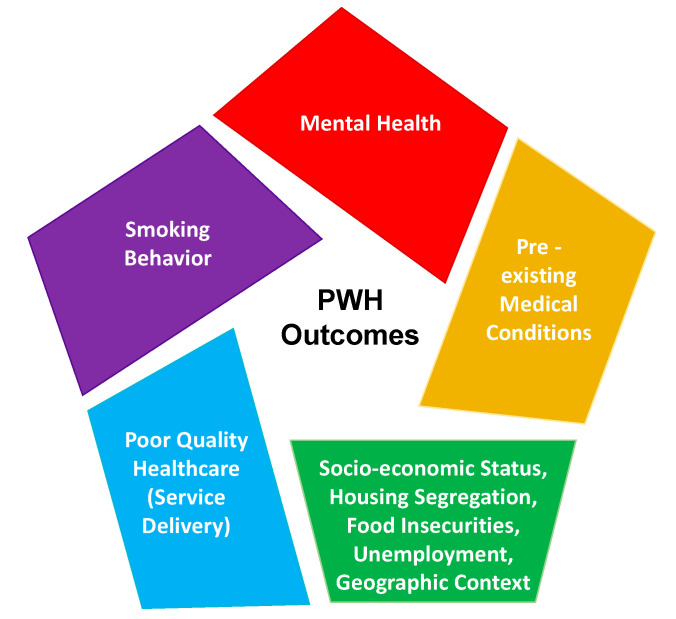
This figure illustrates the syndemic structure of PWH exposures and health outcomes, which needs consideration in addressing comorbid conditions in PWH. It is paramount to have a multilevel approach—a “syndemic framework—that can simultaneously address the medical provider (blue), structural/non-medical (green), and medical (yellow, red, and purple) elements. This syndemic framework illustrates that various causal relationships cannot be disentangled from each element individually because it will require a holistic approach to impact PWH outcomes. NB, this figure was adapted from Sampson et al. [101].

In its application to HIV/AIDS, mental health screening (e.g., depression and anxiety) among PWH should be incorporated into the HIV treatment cascade, and follow-up evaluations for the mental health symptoms should occur frequently at clinical visits [102]. cART can also cause mood changes and anxiety, contributing to the higher prevalence of depression among PWH [103,104,105]. In addition, the physical ailments associated with HIV can have a negative effect on everyday mood, which can keep an individual in a state of depression, as many describe the symptoms as a constant reminder of their infection [97]. Moreover, the economic burden associated with comorbid conditions and the accompanying healthcare costs can threaten the financial stability of individuals, families, and communities and can cause significant stress for PWH. For example, for an individual living with HIV with two or more comorbidities, the costs for transportation to clinic visits and other medical costs are nearly triple that of those without HIV [106]. In LMICs, the economic burden of comorbidities impacts not only the patients, but also their family members and caretakers. The time commitment required of family members to care for PWH often results in lost wages and reduced productivity, both of which can have detrimental effects on the larger economy [97].

## 2. Conclusions

More people live with HIV in Africa than on any other continent. Although many countries have stabilized their HIV epidemics, curtailing the high burden of new HIV infections in many African countries (including South Africa, Nigeria, Uganda, Swaziland, Lesotho, and Zimbabwe) remains a challenge. Moreover, little is known about the number of PWH with comorbidities, including smoking-related conditions in Africa, even as tobacco use continues to rise [107,108]. There is an urgent need for both epidemiological data and the application of implementation research strategies with a focus on health equity to help researchers, policymakers, and healthcare workers understand and address this emerging syndemic. Prospective cohort studies are needed to characterize the dynamic relationship between smoking, biological markers of inflammation, and resultant disease progression in the context of cART use in Africa. Implementation research can be used to adapt strategies to deliver evidence-based smoking cessation interventions shown to be effective in other settings. These parallel research programs will benefit from adopting a syndemic framework to decipher and disentangle the social, behavioral, and biological linkages among HIV, smoking, and cardiopulmonary disease in Africa [97].

## Data Availability

Not applicable.

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
