# Peer review of "An Emerging Syndemic of Smoking and Cardiopulmonary Diseases in People Living with HIV in Africa"

_ijerph, 2021, doi:10.3390/ijerph18063111_

Round 1

Reviewer 1 Report

Emmanuel Peprah and cols. Show an interesting Commentary manuscript regarding Syndemic of Smoking and Cardiopulmonary Diseases in People Living with HIV in Africa; however, some concerns should be resolved.

In general, the manuscript is a little bit disorganized and includes excess information that is not directly relevant.

I have a problem with the type of manuscript selected, in terms of extension and deep their manuscript is a Review.

In the abstract, the authors employ the “tobacco use” term, which is vague; this should be replaced for tobacco smoking, as in the rest of the text.

The references are in number and brackets in the text, but these are ordered by the author, making it hard to follow in the references list.

The authors describe N/A (not apply) in the methods section; I suggest to the authors include the search strategy for papers and statistical reports consulted.

Since results and discussion sections are not appropriate for this manuscript type, they should be deleted and tables and figure 1 relocated.

Reviewer 2 Report

This paper addressed important public health issue of global interest. It is a commentary well articulated with supporting data.

I have comments for authors

  1. Method: It has been written in the abstract but not written in the main text. Please describe the method. It can help future researchers learn how you come up with this paper.
  2. Tables are in the result. What are the sources of the data. If those are citable, please show references. If those tables are not from open access publication, authors should have presented permission of primary authors to reproduce and republish those secondarily. Otherwise It can be of the plagiarism.
  3. The author did not conduct any data collection prospectively. Therefore we understand that those tables are from the references. 

Reviewer 3 Report

This is a commentary on the syndemic of smoking and cardiopulmonary diseases in people living with HIV in Africa.

Major comments

  • The authors need to provide a definition of syndemic, syndemic framework, and how HIV, COPD and CVD can be considered as a syndemic in PWH. More detail is needed about the syndemic framework as it seems to the main topic of the manuscript as shown by the title. There is a lot of detail on smoking, COPD, CAD and inflammation among PWH but it is not clear how it relates to a syndemic or syndemic framework.
  • The authors have provided a figure that illustrates a syndemic structure of PWH exposures and outcomes. This figure has been adapted from Samson et al; however, the figure given in the original paper does not focus on a syndemic framework but illustrates insights and recommendations for reducing health inequities in the United States. I am not sure how this relates to a syndemic framework. Please provide more details and explanations about the figure and how it relates to the syndemic framework.

Round 2

Reviewer 1 Report

Thank you for your kindly and punctual response.

Author Response

We thank reviewer 1 for their kind comments. 

Reviewer 3 Report

The comment about syndemic framework has not been fully addressed by the authors. Although health inequities can be part of a syndemic, the health inequities framework alone cannot explain the syndemic framework. The authors are referred to a paper by Singer et al is for reference and as a guide for a syndemic framework:

Singer M, Bulled N, Ostrach B, Mendenhall E. Syndemics and the biosocial conception of health. Lancet. 2017 Mar 4;389(10072):941-950.
